# The Dynamics of Chromatin Accessibility Prompted by Butyrate-Induced Chromatin Modification in Bovine Cells

Clarissa Boschiero [1], Yahui Gao [1,2], Mei Liu [3], Ransom L. Baldwin VI [1], Li Ma [2], Cong-Jun Li [1,*] and George E. Liu [1,*]

1 Animal Genomics and Improvement Laboratory, Beltsville Agricultural Research Center, Agricultural Research Service, United States Department of Agriculture, Beltsville, MD 20705, USA; clarissa.boschiero@usda.gov (C.B.); gyhalvin@gmail.com (Y.G.); ransom.baldwin@usda.gov (R.L.B.VI)

2 Department of Animal and Avian Sciences, University of Maryland, College Park, MD 20742, USA; lima@umd.edu

3 Animal Nutritional Genome and Germplasm Innovation Research Center, College of Animal Science and Technology, Hunan Agricultural University, Changsha 410128, China; mei.liu@hunau.edu.cn

* Correspondence: congjun.li@usda.gov (C.-J.L.); george.liu@usda.gov (G.E.L.); Tel.: +1-301-504-7216 (C.-J.L.); +1-301-504-9843 (G.E.L.); Fax: +1-301-504-8414 (C.-J.L. & G.E.L.)

**Abstract:** Butyrate is produced by microbe fermentation in the rumen, and its supplementation results in rumen development. Butyrate-induced experiments are valuable in studying nutrient metabolism, cell growth, and functional genomics. This study aimed to characterize accessible chromatin regions and the dynamics of genomic accessibility prompted by butyrate-induced histone modifications in bovine cells. An average of 117,569 accessible chromatin regions were identified for all samples, and 21,347 differentially accessible regions (DARs) for butyrate. Most of the DARs were in distal intergenic regions, introns, and promoters. Gene ontology enrichment resulted in important terms related to the digestive system, regulation of epithelial cells, and cell adhesion. Ingenuity Pathway Analysis (IPA) identified critical networks (e.g., digestive system development, cell morphology and assembly, and cell cycle); canonical pathways (e.g., TGFβ, Integrin-linked kinase, and epithelial adherens junction); and upstream regulators (e.g., TGFβ1, FOS, JUNB, ATF3, and KLFs). Co-expression analysis further showcased the TGFβ and Integrin-linked kinase (ILK)-signaling pathways, which play roles in rumen development through cellular adhesions. This study is the first to provide a genome-wide characterization of differential, open chromatin regions for bovine cells by butyrate-induced treatment. These results provide valuable information for future studies of the butyrate functions in cattle gastrointestinal tract development.

**Keywords:** ATAC-seq; cattle; butyrate; differentially accessible regions; MDBK cells; open chromatin

## 1. Introduction

The key products of microbial fermentation of dietary fiber in the gastrointestinal tract are short-chain fatty acids (SCFAs), including acetate, propionate, and butyrate. Microbial fermentation generates products that contribute to the mammalian energy balance [1,2]. Ruminant species utilize SCFAs to provide up to 70% of their nutrient energy requirements [2]. Among SCFAs, acetate and propionate are the primary energy resources for ruminant metabolism with higher product concentrations. At the same time, butyrate appears to be involved in metabolism not only as a nutrient but also in genome functional regulation [3–5]. Beyond their nutritional importance, SCFAs, especially butyrate, regulate cell differentiation, proliferation, and motility, and induce cell cycle arrest and apoptosis via their histone deacetylases (HDAC) inhibitory activity [6]. The biochemical change in cells treated with butyrate and other HDAC inhibitors is the global hyperacetylation of histones [6–8]. Modifications in chromatin structure are linked to chromosome stability, cell cycle progression, and DNA replication [9]. In addition to in vitro experiments, butyrate supplementation

in preweaning calves stimulates rumen development and accelerates ruminal epithelium maturation [10]. Studies have shown that butyrate supplementation in calves promotes functional rumen development by increasing papillary length, width, and number, and accelerates the oxidation of SCFAs [10,11]. The effect of butyrate supplementation on gene expression has also been reported. For example, a study showed that butyrate infusion on rumen in Holstein cows induced transcriptomic alterations of more than 3500 genes at different timepoints [3].

Chromatin modification has emerged as an important mechanism regulating the genome's transcriptional status [12]. Butyrate effects in bovine cells provide examples of epigenetic regulation and a basis for understanding the butyrate's potential biological roles and molecular mechanisms in animal cell growth and proliferation. Butyrate-induced histone acetylation has a vital role in gene expression [13,14]. However, is still largely unclear how histone modifications are regulated.

ChIP-seq technology was utilized to analyze histone modifications induced by butyrate on a large scale in cattle [13]. Recently, genome-wide epigenomic [15,16] and single-cell transcriptome [17] profiles were generated from bovine rumen epithelial cells before and after butyrate treatment to elucidate the mechanisms of butyrate's role in rumen development. In addition, the first global map of regulatory elements in cattle was generated to evaluate the effect of butyrate treatment, and specific chromatin states were observed after the butyrate treatment [15]. For example, enhancers, a class of cis-regulatory elements, regulate essential processes in animal development. They can also regulate tissue-specific gene expression from over one million base pairs from the promoters in mammals. They can be found within the introns of neighboring genes [18,19]. Chromatin state map studies have identified enhancer states such as active, silent, and poised enhancers. These studies showed that enhancers are the most dynamic genomic portion [20]. However, it is not well understood how enhancers communicate with their target promoters, and their precise mechanisms for functional consequences in chromatin modifications [21], e.g., genome-wide characterization of chromosome accessibility in bovine cells, are still lacking.

Assay of Transposase Accessible Chromatin sequencing (ATAC-seq) is widely used to interrogate chromatin accessibility [22]. Analyzing regulatory elements, such as promoters, enhancers, and insulators, can produce useful biological results. This method uses hyperactive Tn5 transposase, which inserts adapters into accessible regions of chromatin and then reads can be sequenced to obtain regions that are more accessible [23]. ATAC-seq is a sensitive and fast method that can analyze chromatin states, detect chromatin-accessibility signatures and transcription factor (TF) footprints, map nucleosomes, and identify novel enhancers [22,23].

The main objectives of this study were to identify and characterize regions of accessible chromatin in MDBK (Madin–Darby bovine kidney) cells and butyrate-induced treatment using ATAC-seq data to elucidate genetic regulatory elements in bovine cells and the butyrate role in rumen development due to its previously reported effect on rumen development.

## 2. Methods

### 2.1. MDBK Cells and Butyrate Treatment

The Madin–Darby bovine kidney epithelial cells (MDBK; American Type Culture Collection, Manassas, VA; Catalog No. CCL-22) were cultured in Eagle's minimal essential medium and supplemented with 5% fetal bovine serum (Invitrogen, Carlsbad, CA, USA) in 25-cm$^2$ flasks, as described in our previous report [6]. The cells were first seeded and allowed to grow to about 80% confluence. The cells were then reseeded in a 1:5 split ratio (the first passage). The cells used for the experiments were in the second or third passages. At approximately 50% confluence, the cells were treated with 5 mM of sodium butyrate for 24 h during the exponential phase (Calbiochem, San Diego, CA, USA). A butyrate concentration of 5 mM was selected, as it represents a physiologically relevant dose and has previously been successfully used to evoke desired changes in cell cycle dynamics [6].

Two replicate flasks of cells for both butyrate treatment and control groups (four samples in total) were used for the ATAC-sequencing experiments.

### 2.2. ATAC-Seq Library Preparation and Sequencing

ATAC-seq in rumen tissues was performed by Active Motif, Inc. (Carlsbad, CA, USA). The DNA integrity was verified using the Agilent Bioanalyzer 2100 (Agilent, Palo Alto, CA, USA). The DNA was then processed, including end repair, adaptor ligation, and size selection, using an Illumina sample prep kit (Illumina, San Diego, CA, USA) following the manufacturer's instructions. The DNA libraries were then sequenced (75 bp paired-end) on a HiSeq 2500 platform (Illumina, San Diego, CA, USA).

### 2.3. ATAC-Seq Data Processing and Mapping

Sequence reads were first examined for quality using FastQC v.0.11.9 (https://www.bioinformatics.babraham.ac.uk/projects/fastqc/, accessed on 15 June 2021), and adapters and reads with low quality (<20) were removed for all four samples (two biological replicates in each condition). Reads were then aligned to the ARS-USD1.2 cattle reference genome assembly [24] using BWA v.0.7.17 with default settings [25]. Reads that were unmapped, mapped to multiple locations, reads with a mapping quality (MQ) < 10, and reads located on the mitochondrial chromosome were removed by SAMtools v.1.9 [26]. In addition, duplicate reads were removed using Picard v.2.22.3 (https://broadinstitute.github.io/picard/, accessed on 25 June 2021). The fragment size distribution of each sample was created with SAMtools v.1.9 [26].

### 2.4. ATAC-Seq Peak Calling and Quality Check

Individual peaks were called by MACS2 v.2.2.7.1 [27], using the BAMPE parameter (FDR < 0.05) for each sample. Peaks located on chromosome X or unplaced were removed. The fraction of all mapped reads in enriched peaks (FRiP) was obtained for each sample. BEDtools v.2.25.0 [28] Jaccard was used for pairwise comparisons of all samples to obtain the similarity score between samples and the number of peak intersections. In addition, BEDtools v.2.25.0 [28] intersect option was used to merge replicate peaks, and intersect -v option was used to obtain the specific number of peaks for each condition. DiffBind package [29] was used to construct the correlation heatmap using peak information from each sample.

### 2.5. Identification of Differentially Accessible Regions (DARs)

Several steps were applied to obtain a reliable set of differentially accessible regions (DARs). First, DiffReps v.1.55.6 [30] was used to identify the DARs of the butyrate vs. control comparison. A defined window of 200 bp and a G-test (*p*-value < 0.05) were used. BAM files were used as input. Then, the significant differentially accessible regions were defined with an FDR value < 0.01 and log2 fold change $\leq -1$ or log2 fold change $\geq 1$. The last step was to map the significant DARs against the peaks A similar approach was performed before in mice [31]. The identified MAC2 peaks from the four samples (BT1/2, CT1/2) were merged by BEDtools v.2.25.0 [28], generating a list of nonoverlapping peaks. Then, the significant DARs were compared and overlapped against the merged peak list using BEDtools v.2.25.0 [28] with intersect function. In at least one replicate, the DARs that overlapped with MACS2 peaks were further analyzed.

### 2.6. Annotation of DARs

A total of 21,347 unique DARs were annotated with the *annotatePeak* function from the ChIPseeker package [32]. Promoter regions were defined as $\pm 2$ kb from TSS. In addition, the *plotDistToTSS k* function from the ChIPseeker [32] was used to generate the plot of the distribution of transcription factor-binding loci relative to TSS of the DARs. The distance from the regions (binding sites) to the TSS of the nearest gene was calculated by *annotatePeak*.

In addition, butyrate DARs were compared with 15 chromatin states segments previously identified by our group in cattle [15] using the ChromHMM tool [33]. First, all segment coordinates were converted to the ARS-USD1.2 cattle reference genome assembly [24] using liftOver (https://genome.ucsc.edu/cgi-bin/hgLiftOver, accessed on 12 November 2021) with the default parameters (minimum ratio of bases that must remap = 0.95). The converted coordinates were then compared with butyrate DARs using BEDtools v.2.25.0 [28] with intersect function. Then, the enrichment fold of each state was obtained using ChromHMM [33].

### 2.7. Gene Ontology and Pathway Analysis of DARs

Gene Ontology (GO) enrichment analyses were performed using GREAT [34], which predicts functions of cis-regulatory regions and displays enrichment near regulatory elements. First, all coordinates of each DAR were converted to human hg38 using liftOver (https://genome.ucsc.edu/cgi-bin/hgLiftOver, accessed on 20 October 2021) with the default parameters (minMatch = 0.1). Then, the converted coordinates were analyzed using GREAT v.4.0.4 [34] with default parameters. Only results from the hypergeometric test were considered ($p$-value adjusted < 0.05). The GO-Figure tool generated an informative plot with a summary of the GO-enriched terms using semantic similarity (https://gitlab.com/evogenlab/GO-Figure, accessed on 23 October 2021).

QIAGEN Ingenuity Pathway Analysis (IPA) v.68752261 [35] was used with default parameters to find signaling and metabolic pathways of genes with relevant biological functions from butyrate DARs. A total of 7183 unique genes from butyrate DARs were utilized as input for canonical pathways ($p$-value < 0.01), upstream regulators ($p$-value of overlap < 0.01), and molecular networks (network score > 20) analysis.

### 2.8. Motif Enrichment Analysis

A list with 21,347 DARs was used as input to identify enriched motifs in the DARs. Hypergeometric Optimization of Motif EnRichment (HOMER) tool v.4.11 [36] was used with default parameters with *findMotifsGenome* function ($p$-value < 0.01 and >5% of target sequences with motif). HOMER utilizes its curated database of motifs, and most of them are based on published ChIP-Seq data. Each motif has information about cell type, immunoprecipitated protein, and the Gene Expression Omnibus (GEO) accession number or publication information.

In addition, i-cisTarget v.6.0 [37] was used to obtain enriched motifs and predict target genes. First, all coordinates from 21,347 DARs were converted to human hg38 using liftOver (https://genome.ucsc.edu/cgi-bin/hgLiftOver, accessed on 5 October 2021) with the default parameters (minMatch = 0.1). Then, the converted hg38 coordinates were converted again to human hg19 (minMatch = 0.95). A total of 18,666 converted coordinates were used as an input with default parameters. All available databases were included for the analysis, including 24,453 position weight matrices (PWM), 1331 TF binding sites, 2450 histone modifications, and 655 DHS and FAIRE.

### 2.9. Gene Co-Expression Network Analysis

Previous butyrate RNA-seq data (six samples with three biological replicates) was utilized [15] to investigate gene co-expression and regulatory networks and compare with our results. RNA-seq data are available in the NCBI's Gene Expression Omnibus database (accession number GSE129423) [15]. RNA-seq clean reads (Q > 20) were aligned to the ARS-USD1.2 cattle reference genome assembly [24] using STAR v.2.7 [38], and gene expressions were obtained using Cufflinks v.2.2.1 [39]. The FPKM value of each gene was utilized for the weighted correlation network analysis with WGCNA v.1.70-3 [40]. Genes with low expression values for most of the samples were removed before the analyses. Briefly, the topological overlap matrix (TOM) was constructed with soft-thresholding power set to 9, followed by the calculation of corresponding dissimilarity (1-TOM). Then, the identification of modules was performed through the method of dynamic tree cut (minimum size of 20).

Modules whose expression profiles were very similar were further merged by calculating the dissimilarity of module eigengenes (MEs), and color was assigned for each gene module. For module grouping, a threshold of 0.2 was used and corresponds to a correlation of 0.8. The network of genes from selected pathways and co-expressed genes was constructed using VisANT v.5 [41].

## 3. Results

### 3.1. Read Quality, Alignment, and Fragment Size Distribution

Four ATAC-seq libraries were obtained from MDBK cells treated with butyrate (BT, n = 2) and control (CT, n = 2) sequenced on an Illumina HiSeq 2500 platform. A total of 374,049,826 paired-end reads (2 × 75 bp) were generated for all samples with an average of 93,512,457, and the sample CT1 presented the lowest number of reads (Table 1). The fragment size distribution of these ATAC-seq reads was created for each sample from the BAM files (Supplementary File S1). Supplementary File S1 shows that all the samples exhibited the expected fragment sizes with abundant nucleosome-free fragments (<100 bp) and mononucleosomal-spanning fragments. Approximately 98% of the reads were aligned to the ARS-USD1.2 cattle reference genome assembly [24], with a total of 366,935,253 reads mapped and an average of 91,733,813 reads (Table 1). On average, 10.31% of the reads were mapped to the mitochondrial genome; 5.65% were duplicated, and 14.81% had a MQ < 10. A total of 253,399,991 clean reads were produced for all samples (Table 1).

**Table 1.** Sequence read statistics showing the total number of reads, number and percentages of reads mapped, mitochondrial reads, duplicate reads and reads with mapping quality < 10, and the number of clean reads used for peak calling.

| Condition | N of Reads | N of Reads Mapped | % of Mapped Reads | N of MT Reads | % of MT Reads [1] | N of Duplicate Reads | % of Duplicate Reads [1] | N of MQ < 10 Reads | % of MQ < 10 Reads [1] | N of Clean Reads [2] |
|---|---|---|---|---|---|---|---|---|---|---|
| Butyrate 1 | 105,966,982 | 103,703,996 | 97.86 | 8,407,564 | 8.11 | 6,046,975 | 5.83 | 18,580,667 | 17.92 | 71,028,846 |
| Butyrate 2 | 103,071,942 | 100,623,739 | 97.62 | 11,467,359 | 11.40 | 5,539,727 | 5.51 | 17,727,806 | 17.62 | 66,596,923 |
| Control 1 | 54,275,704 | 54,019,636 | 99.53 | 7,069,259 | 13.09 | 1,670,430 | 3.09 | 5,993,034 | 11.09 | 38,580,543 |
| Control 2 | 110,735,198 | 108,587,882 | 98.06 | 9,386,819 | 8.64 | 8,873,305 | 8.17 | 13,695,222 | 12.61 | 77,193,679 |
| Total | 374,049,826 | 366,935,253 | - | 36,331,001 | - | 22,130,437 | - | 55,996,729 | - | 253,399,991 |
| Average | 93,512,457 | 91,733,813 | 98.27 | 9,082,750 | 10.31 | 5,532,609 | 5.65 | 13,999,182 | 14.81 | 63,349,998 |

Each condition has two biological replicates. [1] Percentages were calculated based on the total number of mapped reads. [2] Reads uniquely mapped, with MQ > 10, no duplicate reads, or reads located on MT chromosome.

### 3.2. Identification of Accessible Chromatin Regions

The accessible chromatin regions were identified by the MACS2 tool [27] in all individual samples (FDR < 0.05). A total of 253,399,991 clean reads were used for peak calling (Table 2). In total, 470,274 peaks were identified for all samples, with an average number of 17,569 and an average peak length of 386 (Table 2).

**Table 2.** Peak calling metrics showing the total number of clean reads used to call peaks; the number of clean reads used for a fraction of reads in peaks (FRiP) calculation, number of peaks from MACS2 (FDR < 0.05), number of assigned reads in peaks, FRiP, an average of peak lengths, and proportion of peaks near TSS (±3 Kb, %). Each condition has two biological replicates.

| Condition | N of Clean Reads [1] | N of Clean Reads Used for FRiP [2] | N of MACS2 Peaks [2] | N of Assigned Reads in Peaks [2] | FRiP [3] | Average Peak Length | Proportion of Peaks near TSS (±3 kb, %) |
|---|---|---|---|---|---|---|---|
| Butyrate 1 | 71,028,846 | 69,244,296 | 118,521 | 27,738,114 | 0.39 | 389 | 12.42 |
| Butyrate 2 | 66,596,923 | 64,917,875 | 113,935 | 25,700,148 | 0.39 | 379 | 12.82 |
| Control 1 | 38,580,543 | 37,715,866 | 102,063 | 18,628,115 | 0.48 | 396 | 14.22 |
| Control 2 | 77,193,679 | 75,492,156 | 135,755 | 42,954,062 | 0.56 | 380 | 12.30 |
| Total | 253,399,991 | 247,370,193 | 470,274 | 115,020,439 | - | - | - |
| Average | 63,349,998 | 61,842,548 | 117,569 | 28,755,110 | 0.45 | 386 | 12.94 |

[1] Reads uniquely mapped, with MQ > 10, no duplicate reads, or reads located on MT chromosome. [2] Reads located on chromosomes X and unplaced were not included. [3] Fraction of reads in peaks.

The chromosomal distribution of peaks was obtained for each sample. The distribution was similar among all four samples, with more peaks on chromosomes 1–3, 5, 11, and 19 with an average of >5800 peaks for each chromosome (Supplementary File S2). There were slightly more peaks in the control samples (237,818) than those in the butyrate-treated samples (232,456). The CT1 sample had the lowest number of peaks, with 102,063 peaks. In addition, a specific number of accessible chromatin regions were obtained for each condition. A total of 18,605 butyrate-specific accessible chromatin regions and 21,102 specific accessible chromatin regions for control were identified.

Quality control was performed to verify the quality of the peaks. The heatmap profile of peaks relative to transcription start sites (TSS) $\pm$ 3 kb regions for each replicate can be seen in Figure 1 and shows that the data have a good quality due to the enrichment close to the TSS. The fraction of reads in peaks (FRiP) was obtained for each sample to measure the ATAC-seq quality. The average FRiP for all samples was 0.45. Butyrate samples presented a FRiP of 0.39 and control samples between 0.48–0.56 (Table 2).

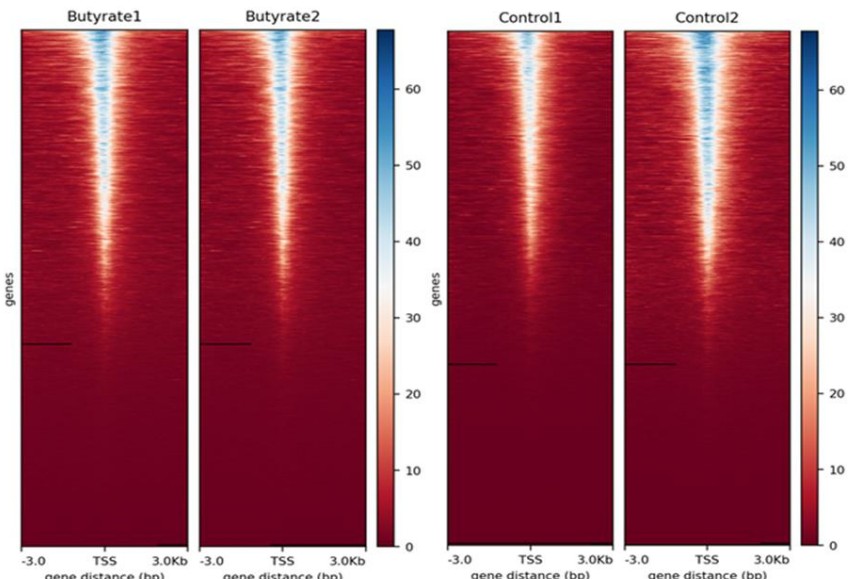

**Figure 1.** Heatmap profile of peaks relative to the transcription start sites (TSS) considering $\pm$ 3 kb regions for each replicate in the butyrate conditions (considering chromosomes 1-29). Blue color intensity reflects the level of peak enrichment. Each condition has two biological replicates.

The correlation heatmap was obtained by the DiffBind package [29] using the data from the peaks for each sample, revealing a clustering of the samples. Figure 2 shows that the biological replicates had high correlations, showing appropriately two distinct clusters of samples (butyrate and control). In addition, the Jaccard similarity index was obtained to measure the similarity of open chromatin regions between two samples, representing the ratio of the number of base pairs in the intersection to the number of base pairs in the union. Jaccard similarity index ranges from 0.0 to 1.0, where 0.0 represents no overlap and 1.0 represents complete overlap. The Jaccard scores for butyrate and control samples were >0.5, and all biological replicates showed similarity (>0.6) (Supplementary File S3).

### 3.3. Differentially Accessible Regions (DARs)

Several steps were applied to obtain a reliable set of DARs. An initial total of 95,629 DARs (*p*-value < 0.05) was obtained for butyrate $\times$ control comparison (Table 3) using the DiffReps tool [30]. Then, the DARs were filtered based on FDR < 0.01 and log2 fold change $\leq -1$ or log2 fold change $\geq 1$, and approximately 24% of the DARs were retained with a total of 22,746 significant DARs (Table 3).

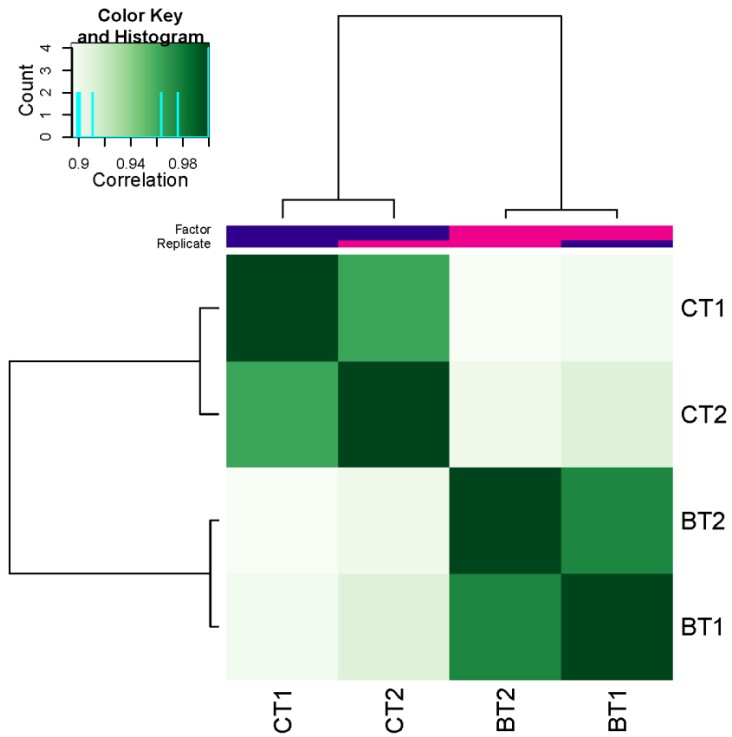

**Figure 2.** Correlation heatmap plot of replicates in the butyrate conditions. Each condition has two biological replicates. BT: Butyrate; CT: Control.

**Table 3.** Number of differentially accessible regions (DARs) for butyrate vs. control comparison, number of significant DARs (FDR < 0.01 and $-1 \leq$ log2FC $\geq 1$), number of significant DARs that overlapped with peaks, and number of unique significant DARs that overlapped with peaks, including induced and repressed DARs.

| Butyrate × Control DARs | N of DARs | % of DARs |
|---|---|---|
| DARs initially identified (*p*-value < 0.05) | 95,629 | - |
| Significant DARs (FDR < 0.01 and $-1 \leq$ log2FC $\geq 1$) | 22,746 | 23.79 |
| Significant DARs that overlapped with peaks | 21,530 | 22.51 |
| Unique significant DARs that overlapped with peaks | 21,347 | - |
| Induced DARs with log2FC $\geq 1$ | 6312 | 29.57 |
| Repressed DARs with log2FC $\leq -1$ | 15,035 | 70.43 |

Then, the 22,746 significant DARs were mapped against the identified MACS2 peaks to obtain a reliable set of significant DARs. DiffReps [30] does not utilize as input the peaks generated by MACS2 [27]. The identified peaks were merged into a single list of 168,742 combined peaks for butyrate (Supplementary File S4), covering 2.78% of the cattle genome (Supplementary File S5). Then, the DARs were compared and overlapped against the merged list of peaks.

Most of the DARs were mapped in the merged peaks in at least one replicate (Supplementary File S4), totaling 21,530 DARs (Table 3, Supplementary File S4). A total of 21,347 unique DARs that overlapped with MACS2 peaks were further annotated and analyzed for enrichment analysis, pathway, and motif enrichment. From the 21,347 DARs, ~70% were classified as repressed DARs, and ~30% were induced DARs (Table 3).

*3.4. Annotation of Differentially Accessible Regions*

Approximately 7% of DARs were in promoter regions with a total of 1529 DARs (Table 4, Supplementary File S6). The majority of the DARs were located in distal intergenic regions (66.5%), introns (23.4%), and promoters (7.16%) (Table 4).

**Table 4.** Annotation of differentially accessible regions (DARs) for butyrate.

| Feature | Number | Frequency (%) |
|---|---|---|
| Promoter (<1 kb) | 1101 | 5.16 |
| Promoter (1–2 kb) | 428 | 2.00 |
| 5′ UTR | 12 | 0.06 |
| 3′ UTR | 118 | 0.55 |
| First Exon | 1 | 0.005 |
| Other Exon | 328 | 1.54 |
| First Intron | 1645 | 7.71 |
| Other Intron | 3357 | 15.73 |
| Downstream (<1 kb) | 47 | 0.22 |
| Downstream (1–2 kb) | 42 | 0.20 |
| Downstream (2–3 kb) | 64 | 0.30 |
| Distal Intergenic | 14,204 | 66.54 |
| Total | 21,347 | 100.00 |

DARs were compared to a previous study that characterized chromatin states in butyrate treatment in rumen epithelial primary cells in cattle [15]. The segments of 647,496 (butyrate), 572,312 (control), and 21,347 butyrate DARs were then compared to the 15 different chromatin state regions, which were successfully converted from the UMD3.1.1 [42] to the ARS-USD1.2 assembly [24] by LiftOver. The majority of the DARs were located on enhancer-related states (EnhA, EnhAATAC, EnhWk, EnhPois, EnhPoisATAC, and EnhWkCTCFATAC) on butyrate (65.22%) or control segments (62.33%), followed by active TSSs (TssA and TssAFlnk) on butyrate (9.57%) or control (9.75%) segments, and one state associated with actively transcribed genes (TxFlnk) on butyrate (5.51%) or control (8.77%) (Supplementary File S7). In addition, the distribution of DARs relative to TSS was obtained (Figure 3). The majority of the DARs in the butyrate condition fall in 10–100 kb regions around the TSS.

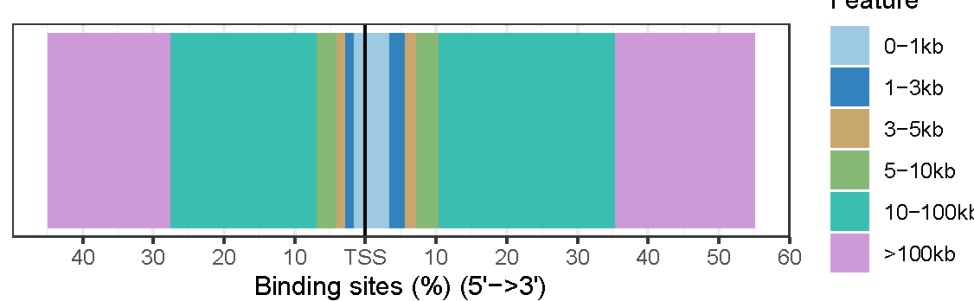

**Figure 3.** Distribution of the butyrate differentially accessible regions (DARs) relative to TSS.

*3.5. Gene Ontology Enrichment Analysis of Differentially Accessible Regions*

DARs from butyrate were utilized for GO enrichment analysis. From the 21,347 unique DARs, ~77% were converted to hg38 (n = 18,683) and used as input for the analysis using GREAT [34]. A total of 238 enriched GO BP, 56 MF, and 55 CC terms were identified (*p*-value adjusted < 0.05) (Supplementary File S8). Three significantly enriched GO terms related to digestive development were identified for butyrate: digestive tract development (GO:0048565), digestive tract morphogenesis (GO:0048546), and digestive system development (GO:0055123). Furthermore, 15 significantly enriched GO terms were related to cell adhesion (regulation of cell adhesion, positive regulation of cell adhesion, regulation of cell–cell adhesion, regulation of cell-substrate adhesion, and others). In addition, 12 significantly enriched GO terms related to the regulation of epithelial cells were found: morphogenesis of an epithelium (GO:0002009), epithelial tube morphogenesis (GO:0060562), morphogenesis of a branching epithelium (GO:0061138), kidney epithelium development (GO:0072073), branching morphogenesis of an epithelial tube (GO:0048754), regulation of epithelial cell migration (GO:0010632), epithelial cell development (GO:0002064), positive regula-

tion of epithelial cell migration (GO:0010634), regulation of epithelial cell proliferation (GO:0050678), columnar/cuboidal epithelial cell differentiation (GO:0002065), positive regulation of epithelial cell proliferation (GO:0050679), and morphogenesis of embryonic epithelium (GO:0016331). Other important, significantly enriched GO terms were cell proliferation, tube development, cell motility, regulation of kinase activity, and response to growth factors (Supplementary File S8).

Furthermore, an informative summary of the GO-enriched terms using semantic similarity to facilitate the interpretation of GOs for butyrate was plotted by the GO-Figure tool (Supplementary File S9). Interesting terms were grouped in anatomical structure formation involved in morphogenesis, tube development, animal organ morphogenesis, blood vessel development, and circulatory system development (Supplementary File S9A); regulation of anatomical structure morphogenesis, regulation of cellular component movement, regulation of cell adhesion, and regulation of locomotion (Supplementary File S9A); kinase binding and cell adhesion molecule binding (Supplementary File S9B); and anchoring junction, membrane raft, actomyosin, and myofibril (Supplementary File S9C).

### 3.6. Pathway Analysis of Differentially Accessible Regions

A total of 7,183 genes from butyrate DARs was used for IPA [35]. Twenty-one significant networks (network score > 20) were identified that related to several essential biological functions, including amino acid metabolism, molecular transport, small molecule biochemistry, cell morphology, cellular function and maintenance, cellular assembly, and organization; cellular compromise, cellular function, and maintenance; digestive system development and function, embryonic development, post-translational modification; cell morphology, cellular function, and maintenance, post-translational modification; cell cycle, cellular movement, connective tissue development and function; and cellular assembly and organization, cellular function and maintenance, and cellular movement (Supplementary File S10).

For canonical pathway analysis, 287 significant pathways (*p*-value < 0.01) were identified, such as TGFβ-signaling, Integrin-linked kinase (ILK)-signaling, Integrin-signaling, epithelial adherens junction-signaling, remodeling of epithelial adherens junctions, regulation of the epithelial–mesenchymal transition pathway, and others (Supplementary File S11). For upstream regulators, 1,513 significant regulators (*p*-value of the overlap < 0.01) were identified, such as TGFβ1, VEGFA, HGF, and other growth factors, and important transcription regulators like FOS, JUNB, ATF3, KLFs (3, 4, 5, 6, and 10), and others (Supplementary File S12).

### 3.7. Motif Enrichment Analysis of Differentially Accessible Regions

HOMER tool [36] was used to identify enriched motifs in the butyrate DARs. A total of 155 enriched motifs (*p*-value ≤ 0.01 and ≥5% of target sequences with motif) were identified (Supplementary File S13). The top ten enriched motifs were: FOS, FRA1, ATF3, BATF, FRA2, AP1, JUNB, FOSL2, JUN-AP1, and CTCF (Supplementary File S13). Eight TFs previously identified in cattle rumen tissue during weaning [43] were also identified by HOMER analysis—ATF3, EGR1, ETS1, FOS, GATA2, JUNB, KLF4, and KLF10 (Supplementary File S13).

In addition to HOMER analysis, enriched motifs and their candidate targets were also identified with the i-cisTarget tool [37]. Results included the Normalized Enrichment Score (NES), the Area Under the Curve (AUC) score, normalized by subtracting the mean of all AUC overall motifs and dividing it by the standard deviation for possible TFs, and candidate targets. The top 10 enriched motifs detected were: JUNB, FRA1, FOSB, FOSL2, FOSL1 (TRANSFAC), JUND (HOCOMOCO), JUN, FOSL1 (HOCOMOCO), JUND (TRANSFAC), and FOS (Figure 4, Supplementary File S14). Eleven TFs previously identified in a study of weaning in cattle rumen tissue [43] were also identified by i-cisTarget analysis—ATF3, EP300, EZH2, FOS, FOSB, GATA2, JUNB, JUND, POLR2A, SMARCA4, and SREBF2.

In the IPA upstream regulators discovery (Supplementary File S12), the same TFs were identified compared to the HOMER motif enrichment analysis, such as ATF3, FOS, GATA1/2/4, JUNB, KLFs (3, 4, 5, 6, and 10), and others (Supplementary File S15).

| Rank | Possible TF | Normalized Enrichment Score (NES) | Number of candidate targets | Database | Logos |
|------|-------------|-----------------------------------|-----------------------------|----------|-------|
| 1 | JUNB | 4.621 | 1482 | PWMs | hocomoco__JUNB_MOUSE.H11MO.0.A |
| 2 | FRA1 | 4.545 | 1509 | PWMs | homer__NNATGASTCATH_Fra1 |
| 3 | FOSB | 4.499 | 1524 | PWMs | transfac_pro__M08936 |
| 4 | FOSL2 | 4.426 | 1484 | PWMs | transfac_pro__M08940 |
| 5 | FOSL1 | 4.425 | 1490 | PWMs | transfac_pro__M08938 |
| 6 | JUND | 4.407 | 1473 | PWMs | hocomoco__JUND_MOUSE.H11MO.0.A |
| 7 | JUN | 4.406 | 1511 | PWMs | transfac_pro__M08942 |
| 8 | FOSL1 | 4.405 | 1478 | PWMs | hocomoco__FOSL1_HUMAN.H11MO.0.A |
| 9 | JUND | 4.397 | 1490 | PWMs | transfac_pro__M08933 |
| 10 | FOS | 4.394 | 1486 | PWMs | dbcorrdb__FOS__ENCSR000DOP_1__m1 |

**Figure 4.** Top 10 motif enrichment discovery results on butyrate differentially accessible regions, including TFs, target genes, and logos.

### 3.8. Co-Expression and Network Visualization of Critical Pathways for Rumen Development

Gene co-expression analysis was conducted to validate essential IPA pathways and construct informative networks. A total of 17,504 genes were utilized for the weighted gene co-expression network analysis (WGCNA) [40]. WGCNA generated 39 merged modules, which ranged from 24 to 9286 genes per module (Supplementary File S16).

Two critical pathways of biological relevance in rumen development were selected—TGFβ-signaling and Integrin-linked kinase (ILK)-signaling (Supplementary File S11). Co-expression information and genes from each significant canonical pathway selected were utilized to construct the networks. TGFβ-signaling pathway was chosen due to its potential role in the cellular adhesions [44,45]. A previous study defined TGFβ as an epithelial cell marker gene in cattle [43]. A total of 58 genes annotated in DARs in butyrate are part of the TGFβ pathway, including *TGFB2*, *TGFB3*, *TGFBR1*, *TGFBR2*, *FOS*, *JUN*, *NODAL SMADs*, and others (Figure 5, Supplementary File S11). In addition, canonical pathway results detected TGFβ genes as part of other pathways (Supplementary File S11). The *TGFB2* was selected as the hub gene for the network. The *TGFB1* gene was not included because no differentially accessible chromatin region was identified near this gene. Twenty of the

fifty-eight genes present in the network showed a high co-expression (>0.8) with *TGFB2* (Figure 5). Another vital pathway selected was the Integrin-linked kinase (ILK)-signaling (Figure 6). The ILK-signaling pathway was also chosen due to its potential role in cellular adhesions in mammals [46,47]. The *ILK* gene was selected as the hub gene for the network. Of the 102 genes present in the network, 60 showed a high co-expression (>0.8) with *ILK*.

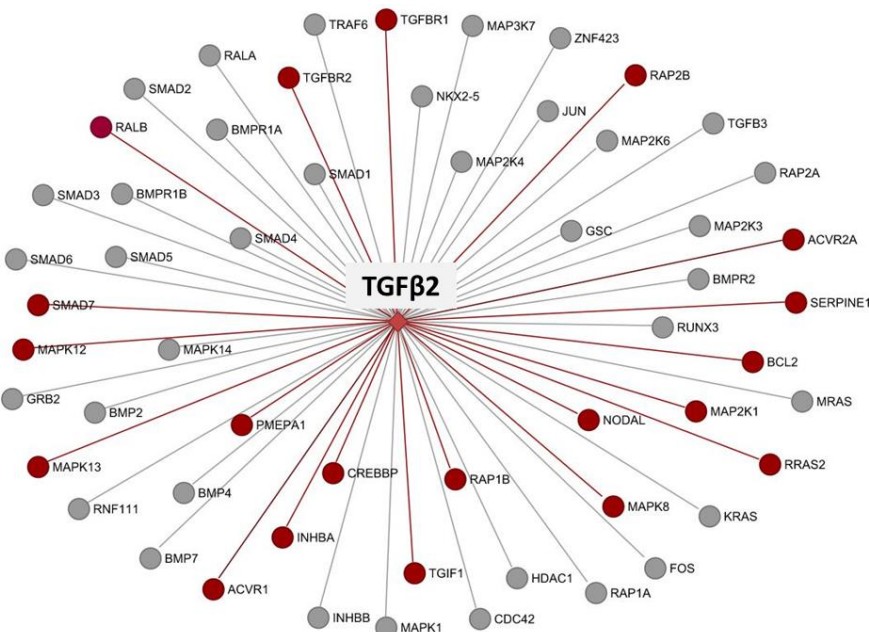

**Figure 5.** TGFβ-signaling pathway network. Genes included are from IPA canonical pathway, and all of them are in differentially accessible regions (DARs) in butyrate. Red edges represent significant co-expressed genes (>0.8) using RNA-seq data. The *TGFB2* was selected as the hub gene for the network.

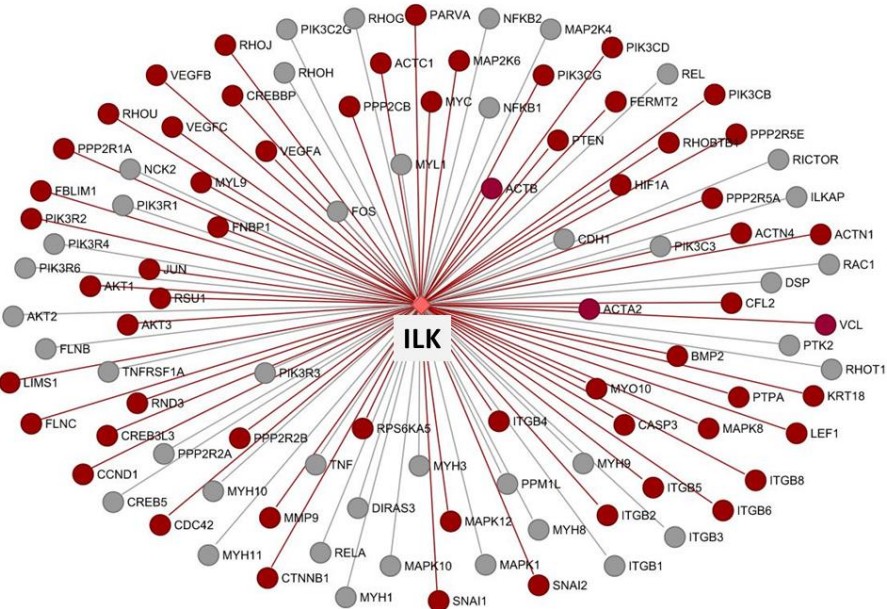

**Figure 6.** Integrin-linked kinase (ILK)-signaling pathway network. Genes included are from IPA canonical pathway, and all of them are in differentially accessible regions (DARs) in butyrate. Red edges represent significant co-expressed genes (>0.8) using RNA-seq data. The *ILK* was selected as the hub gene for the network.

## 4. Discussion

After the significant achievement of genomic sequencing and mapping projects, a similarly substantial challenge is to functionally annotate the animal genome, especially regulatory elements. DNA is packed within a five-micron nucleus in a mammal cell, with the hierarchical folding of DNA around core histone proteins to form nucleosomes. Millions of nucleosomes then compact into chromatin. This hierarchical packaging segregates inactive genomic regions. The promoters, enhancers, or other regulatory elements of active biological regions are open and accessible to the transcription machinery [48]. Therefore, functional regulatory elements can be characterized by defining and measuring chromatin accessibility.

The bovine genome still has a large, uncharacterized noncoding portion, especially those relating to regulatory elements. Active regulatory elements can be characterized by chromatin accessibility, such as the ATAC-seq approach [22]. Until today, few studies have used ATAC-seq technology in cattle tissues to describe regions of open chromatin, such as in the bronchial lymph nodes of dairy calves [49]; the liver, muscle, and hypothalamus of indicine cattle [50]; muscle from Qinchuan cattle [51]; and eight adult tissues (adipose, cerebellum, brain cortex, hypothalamus, liver, lung, muscle, and spleen) from Hereford cattle [52]. An additional study utilized ATAC-seq and other technologies to annotate and identify chromatin states in rumen cattle tissue under the butyrate treatment [15].

In this study, the ATAC-seq approach was used to identify genomic regulatory elements in bovine cells (MDBK) treated with butyrate. Distinct chromatin accessibility profiles were determined at different conditions, indicating the importance of the regulatory effect of the butyrate treatment in bovine cells. However, one of the limitations of this study was the small sample size utilized (n = 2), although two biological replicates were used for each sample. Another limitation was the use of Madin–Darby bovine kidney epithelial cells (MDBK) instead of rumen epithelial primary cells (REPC) or rumen tissue. A previous study of butyrate-induced treatment in bovine cells and rumen tissue reported consistent chromatin states identification in REPC, MDBK, and rumen tissue [15]. In addition, the authors verified that cell lines can be used for butyrate-induced studies to verify chromatin states.

Butyrate samples resulted in an average of 63,349,998 clean reads. The ENCODE ATAC-seq pipeline recommends 50 million reads for paired-end sequencing for each replicate (https://www.encodeproject.org/atac-seq/, accessed on 8 November 2021), and this quality standard was met for most samples. To verify the quality of the peaks in each biological replicate, the fraction of reads in peaks (FRiP score), the Jaccard similarity index, and a correlation heatmap were obtained. The ENCODE project recommends a minimum FRiP score of 0.2 for ATAC-seq libraries (https://www.encodeproject.org/atac-seq/, accessed on 8 November 2021). In this study, the minimum FRiP score was 0.39 for BT1 and BT2. In addition, the replicate correlation showed that all replicates in this study are highly correlated and appropriate to be used in the downstream analysis. In addition, the Jaccard similarity index corroborated with the replicate correlation, showing that replicates have a remarkable similarity (Jaccard score of >0.6). A previous study in ATAC-seq in cattle also used the Jaccard score to measure similarity between samples and check their quality [49].

An average of 117,569 peaks were identified for all samples. There were slightly more regions of accessible chromatin identified in the control samples than in the butyrate samples, despite the sample CT1 having the lowest number of peaks of all samples. The low number of peaks in the CT1 sample is probably explained by its low number of reads. Then, by merging all peaks from the four samples, a total of 168,742 combined peaks was obtained, including shared and condition-specific regions, covering 2.78% of the cattle genome. Previous studies identified a similar number of peaks in each tissue/replicate, ranging from ~37,000–249,000 peaks [52], ~22,000–48,000 peaks [49], and 19,000–25,000 peaks [51]. Another study on cattle reported consensus peaks for different tissues, ranging from ~22,000 to 78,000 peaks [50].

To obtain the DARs, we utilized DiffReps [30]. DiffReps software has a sliding window approach to scan the genome for enrichment regions and provides different statistical tests, such as chi-square and G-test, to detect differential chromatin sites with or without biological replicates [30]. The G-test performs a goodness-of-fit test on normalized counts. Results indicated that DiffReps is an accurate software to detect differential sites from ChIP-seq data [30]. A total of 21,347 DARs were discovered in the butyrate vs. control comparison. A recent cattle study comparing adult and embryo muscle tissues identified 8850 DARs [51]. All these results indicate that our ATAC-seq data were high quality and informative.

Annotation of the DARs revealed that the majority were in distal intergenic, introns, and promoter regions, indicating that distal regulatory elements play an essential role in the butyrate condition. By summarizing the distances from a DAR to its nearest TSS, it was also observed that most of the DARs fall in 10–100 kb regions away from the TSS, again suggesting that these ATAC-seq sites could be distal enhancers. DARs were also compared to a previous study that identified chromatin states in REPC, MDBK, and bovine rumen tissue [15]. This study revealed that weak enhancers and flanking active transcriptional start sites were the most dynamic states [15]. Butyrate can induce essential modifications of the epigenomics landscape in cattle. Interestingly, overlapping DARs with 15 chromatin states previously identified in cattle rumen tissue and cell lines [15] revealed that most of the DARs were located on enhancer states.

Gene ontology enrichment results revealed important, significantly enriched GO terms related to the digestive system, regulation of epithelial cells, cell adhesion and proliferation, cell motility, tube development, and regulation of kinase activity. In addition, the semantic similarity of GO terms was obtained. The scatterplots produced by the GO-Figure software reduce the complete list of terms by capturing the main biological features from the results. For biological processes, the main processes were associated with three groups, one related to tube/blood vessel/circulatory system development; a second group related mainly to the regulation of cell adhesion/locomotion/cellular component movement; and a third small group linked to cell migration. Kinase binding and cell adhesion molecule binding were the major molecular function processes, and anchoring junction was the highly significant GO term in the cellular component process. Cell adhesion has a vital role in the development and maintenance of tissues, and this cellular process is essential for regulating the survival, proliferation, migration, and communication of cells [53]. Protein kinases regulate several fundamental biological processes, such as cell growth, signaling and proliferation, regulation of immune responses, and others [54,55].

IPA analyses have also identified essential networks of biological relevance (e.g., digestive system development and function, cell morphology and assembly, cellular function and maintenance, cell cycle, cellular movement, and others); canonical pathways (e.g., TGFβ, Integrin-linked kinase, Integrin, and epithelial adherens junction signaling); and upstream regulators (e.g., TGFβ1, FOS, JUNB, ATF3, KLFs, MAPK1, and SMARCA4).

Two important pathways of biological relevance in rumen development were selected—TGFβ-signaling and ILK-signaling. Previous studies revealed that TGFβ [44,45] and ILK [46,47] have a potential role in cellular adhesions. A study on cattle showed that TGFβ is an epithelial cell marker gene [43]. Furthermore, in a study on sheep fed a fibrous diet [56], different proteins were detected in rumen epithelial cells, such as KRTs, TGFα, and TGFβ1. TGFβ1 has a potential role related to cell growth, metabolism, and cell adhesion in different cell types—sheep rumen epithelial cells [56], cattle rumen epithelial cells [57], and human dermal fibroblasts [58]. Another two studies on cattle showed that TGFβ1 has a potential role in the rumen epithelial development [43,59]. Although TGFβ1 was not identified in an accessible region/differentially accessible region in this study, other related proteins were found, such as TGFβ2, TGFβ3, TGFβR1, TGFβR2, and TGFβI. Our results suggest that not only TGFβ1, but related proteins, such as TGFβ2, may have an essential role in cattle rumen development.

Furthermore, motif enrichment analyses revealed important candidate TFs for rumen development, such as AP1, ATF3, BATF, FOS, FOSB, FOSL1/2, FRA1/2, JUN/B/D, and KLFs (3, 4, 5, 6, and 10). The same TFs were previously identified in a study evaluating two Holstein calf ruminal epithelial tissues before and after weaning [43], such as: ATF3 (HOMER/ i-cisTarget), EGR1 (HOMER), EP300 (i-cisTarget), ETS1 (HOMER), EZH2 (i-cisTarget), FOS (HOMER/ i-cisTarget), FOSB (i-cisTarget), GATA2 (HOMER/ i-cisTarget), JUNB (HOMER/ i-cisTarget), JUND (i-cisTarget), KLF4 (HOMER), KLF10 (HOMER), POLR2A (i-cisTarget), SMARCA4 (i-cisTarget), and SREBF2 (i-cisTarget). Krüppel-like factors (KLFs) are members of the zinc-finger family of TFs and regulate many development processes and activate/repress a large number of genes [60,61]. KLF4 was initially identified as a gut-enriched TF in the intestine [62], but further studies showed its expression in other organs and tissues, such as skin, kidneys, endothelial cells, vascular smooth muscle cells, and others [63]. KLF4 regulates important cellular processes like cell proliferation and differentiation [63], adipogenesis [60], and mesenchymal–epithelial transition induction [64]. *KLF10* is a TGFβ-induced gene and has been implicated in multiple functions such as cell differentiation, apoptosis, osteoblast and osteoclast differentiation, gluconeogenesis, and inflammation [65–67]. A recent study showed that KLF10 is part of the sugar metabolism-related signaling pathway and protects against the negative effects of increased sugar consumption [68]. An evolutionary study on genomic rearrangements in the ruminants [69] identified 25 TFs, including KLF4/5 enriched in liver enhancers near ruminant breakpoint regions, showing that these TFs may have a fundamental role in ruminants. JUN, JUNB, JUND, FRA1, FRA2, FOSB, and ATF3 proteins are all AP1 (activator protein 1) transcription factor families with several biological roles, including proliferation, differentiation, and cell death [70]. JUNB plays an important role in cell proliferation and controls different phases of the cell cycle [71]. JUND activates and represses several target genes and has a crucial role in cell growth [72]. SMARCA4 has a potential role in cell differentiation and mediates important developmental events such as embryonic activation in cattle [73]. GATA1, 2, and 4 were identified in the HOMER enrichment motif and IPA analyses. GATA TFs have a role in cell proliferation and differentiation. In a study on mice, GATA4/6 were identified as potential regulators of cell proliferation and differentiation in the small intestine [74]. In summary, the literature, together with these results, may indicate that transcription factors such as ATF3, FOS/FOSB, GATA1/2/4, FRA1/2, KLFs, JUN/B/D, and SMARCA4 have important roles in rumen development. Furthermore, this study indicates that butyrate-induced treatment in transformed bovine cells results in chromatin modifications and enhancer activations and reveals TFs and candidate target genes that can have important functions and may regulate rumen development.

## 5. Conclusions

By using the butyrate-induced treatment on MDBK cells and the ATAC-seq approach, genome-wide characterization of differential, open chromatin regions and regulatory elements was obtained for the first time in bovine cells. Distinct chromatin accessibility profiles were identified in bovine MDBK cells under butyrate, showing the importance of the regulatory effect of the butyrate. In addition, the identification of DARs, and their possible biological roles, evidenced by gene ontology results, pathways, motif enrichment, and co-expression information, resulted in essential enhancers, transcription factors, and candidate target genes for rumen development. Important enriched GO terms were found related to the digestive system, regulation of epithelial cells, cell adhesion, cell proliferation, tube development, and regulation of kinase activity. Two canonical signaling pathways (TGFβ and ILK) were selected as examples due to their potential role in cellular adhesions and integrated with co-expression information. This study revealed potential candidate genes and TFs for rumen biology in bovine cells that can help scientists better understand butyrate functions. Additional studies with larger sample sizes and in vivo experiments are needed to confirm these results.

**Supplementary Materials:** The following supporting information can be downloaded at: https://www.mdpi.com/article/10.3390/ruminants2020015/s1. Supplementary Files S1–S16.

**Author Contributions:** C.-J.L. and G.E.L. conceived and designed the experiments. C.B., Y.G. and M.L. performed computational and statistical analyses. R.L.B.VI and L.M. provided tissue and computational resources. C.B., G.E.L. and C.-J.L. wrote the paper. All authors have read and agreed to the published version of the manuscript.

**Funding:** This work was supported in part by AFRI grant numbers 2019-67015-29321, 2020-67015-31398, and 2021-67015-33409 from the USDA National Institute of Food and Agriculture (NIFA) Animal Genome and Reproduction Programs.

**Institutional Review Board Statement:** No animal was used in this study and no ethics approval was needed.

**Informed Consent Statement:** Not applicable.

**Data Availability Statement:** All high-throughput sequencing data analyzed in this study are deposited in NCBI. RNA-seq data are publicly available at the NCBI GEO database under accession number GSE129423 [15]. All ATAC-seq data were submitted to NCBI, SRA database (SUB8420017, BioProject ID: PRJNA672996).

**Acknowledgments:** We thank Erin Connor, Reuben Anderson, Mary Bowman, Donald Carbaugh, Christina Clover, Sarah McQueeney, Mary Niland, Marsha Campbell, Dennis Hucht, and Research Animal Services staff at the Beltsville Dairy Unit for technical assistance. The mention of trade names or commercial products in this article is solely to provide specific information and does not imply recommendation or endorsement by the U.S. Department of Agriculture (USDA). The USDA is an equal opportunity provider and employer.

**Conflicts of Interest:** The authors declare that they have no competing interests.

## Abbreviations

| | |
|---|---|
| ATAC-seq | assay for transposase-accessible chromatin sequencing |
| AUC | Area Under the Curve |
| BAM | binary alignment map |
| BP | biological process |
| BT | butyrate |
| CC | cellular component |
| ChIP | chromatin immunoprecipitation |
| CT | control |
| DARs | differentially accessible regions |
| DHS | DNase I hypersensitive sites |
| ENCODE | The Encyclopedia of DNA Elements |
| FAIRE | formaldehyde-assisted isolation of regulatory elements |
| FDR | false discovery rate |
| FRiP | fraction of reads in peaks |
| GEO | Gene Expression Omnibus |
| GO | Gene Ontology |
| HDAC | histone deacetylases |
| ILK | integrin-linked kinase |
| IPA | ingenuity pathway analysis |
| MDBK | Madin–Darby bovine kidney |
| MF | molecular function |
| PWM | position weight matrices |
| REPC | rumen epithelial primary cells |
| SCFAs | short-chain fatty acids |
| TF | transcription factor |
| TOM | topological overlap matrix |
| TSS | transcription starting site |

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
