# Peer review of "The Dynamics of Chromatin Accessibility Prompted by Butyrate-Induced Chromatin Modification in Bovine Cells"

_ruminants, doi:10.3390/ruminants2020015_

Round 1

Reviewer 1 Report

The histone modifications are a very important issue to chromatin accessibility and gene expression, with major benefits for the animal and human biotechnologies. The present study aimed to identify and characterize regions of accessible chromatin in MDBK (Madin–Darby bovine kidney) cells and butyrate-induced chromatin accessibility using ATAC-seq data to elucidate genetic regulatory elements in bovine cells.

I did not correct English sentences; however, there are some grammatical and stylistic corrections that need to be made. Make sure that your references, reagents, commercial kits, equipment are cited throughout the text according to the journal's guidelines.

This manuscript shows that several transcription factors are possible candidates in rumen development and that butyrate treatment alters chromatin stability, enhancer activations. The authors also showed possible candidate target genes that may play an important role in rumen development and regulation. In my opinion, the present manuscript is appropriate for publication in Journal Ruminants.

Author Response

R: Thank you for these observations and suggestions. We improved English writing and grammar, and we also made the necessary modifications for the references of the reagents/equipment cited according to the Journal instructions. We also improved the abstract and reduced its length (200 words) according to the Journal guidelines. We improved the Discussion section by adding the limitations of the study and we also improved the Conclusions section.

Reviewer 2 Report

The authors could explain better the results, its specialized to some lectors, and if they can not are specialists, the relevance of the study is not understanding

Author Response

R: Thanks for pointing that out. We added more relevant details, and we removed redundant information to clarify our text and results as suggested. We improved the Discussion section by adding the limitations of the study and we also improved the Abstract and Conclusions section.

Reviewer 3 Report

In the manuscript 'The dynamics of chromatin accessibility prompted by butyrate-induced chromatin modification in bovine cells,' the authors aimed to identify differences in chromatin accessibility induced by Butyrate exposure using the MDBK cell line. The data presented support the author's hypothesis that butyrate, an SCFA compound produced by ruminal microbes, induces differential accessible chromatin regions.

Overall, this paper reveals altered chromatin accessibility in MDBK cells treated with butyrate, but it is unclear how this relates to rumen development or function. The MDBK cell line is a transformed line that is a generic epithelial cell type, whereas the ruminal epithelium is a specialized stratified epithelium. Further, these cells are likely of advanced passage number and significantly adapted to in vitro culture conditions. Therefore, the authors should interpret these data cautiously. Why did the authors not elect to conduct an in-vivo experiment with calves and a ruminal epithelium primary culture, as this might prove more insightful in understanding the role of butyrate (produced by ruminal microbes) on chromatin architecture and its implications on ruminal development?

In the discussion section, the authors need to expand on the significant limitations of their experiments and acknowledge that correlating chromatin states between the ruminal epithelium and MDBK line is a stretch. 

Other comments:

  • To enhance reproducibility, the authors need to indicate the passage number for the MDBK line.
  • The sequencing parameters, read length, depth, and statistics follow ATAC seq standards.
  • The authors have used an n=2; this seems too small an N, especially for a cell culture experiment.
  • On line 170, the authors filtered their differentially accessible regions (DAR) based on log2fold > 1 after FDR correction. Log2fold > 1 seems a bit low. This parameter needs to be at least 1.5 or maybe 2?
  • Fig.3 shows TF binding sites for the DARs in butyrate. The authors should add a figure for the control sample for a side-by-side comparison.
  • Section 2.5 describes the GO analysis done on the DARs. The authors considered enrichment of p-adjusted < 0.05, it would be helpful to know how low the p values were. Maybe a table or a horizontal bar plot might help. This criticism also applies to pathway analysis.
  • The paragraph on lines 374 - 378 seems out of place and should be moved to better integrate the core ideas. "A previous study of the chromatin states in the cattle rumen [15] revealed that weak enhancers and flanking active transcriptional start sites were the most dynamic states. Butyrate can induce essential modifications of the epigenomics landscape in cattle. Interestingly, overlapping DARs with 15 chromatin states previously identified in cattle rumen tissue [15] revealed that most of the DARs were located on enhancer states. "
  • On line 445" Furthermore, this study indicates that butyrate-induced treatment in cattle results in ….…." The authors should use 'transformed bovine cells' instead of cattle, which implies the authors used living animals.
  • By rumen development, do the authors mean developmental origins of the rumen? Or the development that occurs in calves post-weaning when they start solid feed? The authors should clarify this.

Author Response

Overall, this paper reveals altered chromatin accessibility in MDBK cells treated with butyrate, but it is unclear how this relates to rumen development or function. The MDBK cell line is a transformed line that is a generic epithelial cell type, whereas the ruminal epithelium is a specialized stratified epithelium. Further, these cells are likely of advanced passage number and significantly adapted to in vitro culture conditions. Therefore, the authors should interpret these data cautiously.

Why did the authors not elect to conduct an in-vivo experiment with calves and a ruminal epithelium primary culture, as this might prove more insightful in understanding the role of butyrate (produced by ruminal microbes) on chromatin architecture and its implications on ruminal development?

In the discussion section, the authors need to expand on the significant limitations of their experiments and acknowledge that correlating chromatin states between the ruminal epithelium and MDBK line is a stretch. 

R: Thank you for your suggestion. We understand this suggestion and concern, however a previous study in cattle using cell lines (REC and MDBK) and rumen tissue with butyrate treatment showed consistent results for the identification of chromatin states for all cell lines and rumen tissue (Fang et al., 2019), and they concluded that cell lines are adequate for this type of study. But we agree with the reviewer that, in vivo experiments are necessary for future to validate these results. To clarify this, we added more information about this limitation in lines 348-353, 394-395, 399, and 486-487.

Fang L, Liu S, Liu M, Kang X, Lin S, Li B, Connor EE, Baldwin RL, Tenesa A, Ma L, et al: Functional annotation of the cattle genome through systematic discovery and characterization of chromatin states and butyrate-induced variations. BMC Biol 2019, 17:68.

To enhance reproducibility, the authors need to indicate the passage number for the MDBK line.

R: Thanks for pointing that out and the information is now described in Methods.

The authors have used an n=2; this seems too small an N, especially for a cell culture experiment.

R: Thanks for your suggestion. We understand that the use of only two samples (with 2 biological replicates for each one) is a small sample size, and has limitations, so we included this in the discussion and conclusions sections to clarify the limitations of our study (Lines 346-348; 486-487).

On line 170, the authors filtered their differentially accessible regions (DAR) based on log2fold > 1 after FDR correction. Log2fold > 1 seems a bit low. This parameter needs to be at least 1.5 or maybe 2?

R: Thanks for pointing that out. We carefully tested other log2 fold change cut-offs (1.5 and 2), but it resulted in an insufficient number to conduct an accurate downstream analysis, especially for the induced DARs, for example, with log2 fold change cut-off of 2, only 821 induced DAR were identified.  We strongly believe that the combination of the stringent FDR cut-off of 0.01, log2fold-change > 1, and the overlapping with MACS2 peaks resulted in a high-quality dataset of DARs. A similar approach was performed before in mice (Lodato et al., 2018 and Matthews et al., 2020).

  • Lodato NJ, Rampersaud A, Waxman DJ: Impact of CAR Agonist Ligand TCPOBOP on Mouse Liver Chromatin Accessibility. Toxicol Sci 2018, 164:115-128.

  • Matthews BJ, Waxman DJ. Impact of 3D genome organization, guided by cohesin and CTCF looping, on sex-biased chromatin interactions and gene expression in mouse liver. Epigenetics Chromatin. 2020 Jul 17;13(1):30. doi: 10.1186/s13072-020-00350-y. PMID: 32680543; PMCID: PMC7368777.

Fig.3 shows TF binding sites for the DARs in butyrate. The authors should add a figure for the control sample for a side-by-side comparison.

R: Thanks for your suggestion. We presented only one figure because this figure shows the TF binding sites of the regions that are differentially accessible regions between Butyrate x control (butyrate DARs), so there is no need to present control x butyrate because we did not include this comparison in our downstream analyses and would be redundant. All downstream analyses are from the DARs of butyrate (butyrate x control comparison).

Section 2.5 describes the GO analysis done on the DARs. The authors considered enrichment of p-adjusted < 0.05, it would be helpful to know how low the p values were. Maybe a table or a horizontal bar plot might help. This criticism also applies to pathway analysis.

R: Thanks for your suggestion. We already added a plot visualization of the GO results using the GO-Figure tool (Additional File 9). This tool considered the p-values and functions of the GO enriched terms to group them in a plot. We obtained a list of 238 significant GO terms the for biological process category. It will be difficult to plot all these results together with a good resolution for the audience, and because of this, we choose GO-Figure plots. But, as the reviewer suggested, we added in the Additional File 8, a color scale in the table for the p-values and the fold enrichment to facilitate the visualization of the p-values/fold enrichment. In the canonical pathways, we also added a color scale for the p-values to facilitate the visualization (Additional File 11). In the Additional File 12, we also added the color scale.

The paragraph on lines 374 - 378 seems out of place and should be moved to better integrate the core ideas. "A previous study of the chromatin states in the cattle rumen [15] revealed that weak enhancers and flanking active transcriptional start sites were the most dynamic states. Butyrate can induce essential modifications of the epigenomics landscape in cattle. Interestingly, overlapping DARs with 15 chromatin states previously identified in cattle rumen tissue [15] revealed that most of the DARs were located on enhancer states. "

R: Thanks for your suggestion. We modified the location of this paragraph and merged it with the previous one, and we also modified the text to improve the understanding (lines 387-400).

On line 445" Furthermore, this study indicates that butyrate-induced treatment in cattle results in ….…." The authors should use 'transformed bovine cells' instead of cattle, which implies the authors used living animals.

R: Thanks for pointing that out. We modified it as suggested.

By rumen development, do the authors mean developmental origins of the rumen? Or the development that occurs in calves post-weaning when they start solid feed? The authors should clarify this.

R: Thanks for pointing that out. We mentioned rumen development in the context that exogenous supplementation with butyrate in calves has been shown to accelerate rumen development.  To clarify this, we included an additional reference about the supplementation of butyrate and the rumen development in the Introduction (Review: Exogenous butyrate: implications for the functional development of ruminal epithelium and calf performance) in lines 62-66. And we added more information to clarify this in Abstract (lines 17-18), and Introduction (lines 102-103). We also modified the conclusions to be clearer. 

Round 2

Reviewer 3 Report

The authors have addressed all my concerns.